# EFFICIENT RECURRENT ARCHITECTURES THROUGH ACTIVITY SPARSITY AND SPARSE BACK-PROPAGATION THROUGH TIME

**Anand Subramoney**[1,2,*]**, Khaleelulla Khan Nazeer**[3]**, Mark Schöne**[3]**, Christian Mayr**[3,4]**, David Kappel**[1]

[1] Institute for Neural Computation, Ruhr University Bochum, Germany

[2] Royal Holloway, University of London

[3] Faculty of Electrical and Computer Engineering, Technische Universität Dresden, Dresden, Germany

[4] Centre for Tactile Internet with Human-in-the-Loop (CeTI),Technische Universität Dresden, Dresden, Germany

`{anand.subramoney,david.kappel}@ini.rub.de`

`{khaleelulla.khan_nazeer,mark.schoene,christian.mayr}@tu-dresden.de`

## ABSTRACT

Recurrent neural networks (RNNs) are well suited for solving sequence tasks in resource-constrained systems due to their expressivity and low computational requirements. However, there is still a need to bridge the gap between what RNNs are capable of in terms of efficiency and performance and real-world application requirements. The memory and computational requirements arising from propagating the activations of all the neurons at every time step to every connected neuron, together with the sequential dependence of activations, contribute to the inefficiency of training and using RNNs. We propose a solution inspired by biological neuron dynamics that makes the communication between RNN units sparse and discrete. This makes the backward pass with backpropagation through time (BPTT) computationally sparse and efficient as well. We base our model on the gated recurrent unit (GRU), extending it with units that emit discrete events for communication triggered by a threshold so that no information is communicated to other units in the absence of events. We show theoretically that the communication between units, and hence the computation required for both the forward and back-ward passes, scales with the number of events in the network. Our model achieves efficiency without compromising task performance, demonstrating competitive performance compared to state-of-the-art recurrent network models in real-world tasks, including language modeling. The dynamic activity sparsity mechanism also makes our model well suited for novel energy-efficient neuromorphic hardware. Code is available at https://github.com/KhaleelKhan/EvNN/.

## 1 INTRODUCTION

Large scale models such as GPT-3 (Brown et al., 2020) and DALL-E (Ramesh et al., 2021) have demonstrated that scaling up deep learning models to billions of parameters improve not just their performance but lead to entirely new forms of generalisation. But for resource constrained environments, transformers are impractical due to their computational and memory requirements during training as well as inference. Recurrent neural networks (RNNs) may provide a viable alternative in such low-resource environments, but require further algorithmic and computational optimizations. While it is unknown if scaling up recurrent neural networks can lead to similar forms of generalization, the limitations on scaling them up preclude studying this possibility. The dependence of each time step's computation on the previous time step's output prevents easy parallelisation of the model computation. Moreover, propagating the activations of all the units in each time step is computationally inefficient and leads to high memory requirements when training with backpropagation through time (BPTT).

While allowing extraordinary task performance, the biological brain's recurrent architecture is extremely energy efficient (Mead, 2020). One of the brain's strategies to reach these high levels of efficiency is activity sparsity. In the brain, (asynchronous) event-based and activity-sparse communication results from the properties of the specific physical and biological substrate on which

---

*Work done while at Ruhr University Bochum

the brain is built. Biologically realistic spiking neural networks and neuromorphic hardware aim to use these principles to build energy-efficient software and hardware models (Roy et al., 2019; Schuman et al., 2017). However, despite progress in recent years, their task performance has been relatively limited for real-world tasks compared to recurrent architectures based on LSTM and GRU.

In this work, we propose an activity sparsity mechanism inspired by biological neuron models, to reduce the computation required by RNNs at each time step. Our method adds a mechanism to the recurrent units to emit discrete events for communication triggered by a threshold so that no information is communicated to other units in the absence of events. With event-based communication, units in the model can decide when to send updates to other units, which then trigger the update of receiving units. When events are sent sparingly, this leads to activity-sparsity where most units do not send updates to other units most of the time, leading to substantial computational savings during training and inference. We formulate the gradient updates of the network to be sparse using a novel method, extending the benefit of the computational savings to training time. We theoretically show, in the continuous time limit, that the time complexity of calculating weight updates is proportional to the number of events in the network. We demonstrate these properties using Gated Recurrent Unit (GRU) (Cho et al., 2014) as a case study, and call our model Event-based Gated Recurrent Unit (EGRU). We note, however, that our dynamic activity-sparsity mechanism can be applied to any RNN architecture.

In summary, the main contributions of this paper are the following:

1. We introduce a variant of the GRU with an event-generating mechanism, called the EGRU.

2. We theoretically show that, in the continuous time limit, both the forward pass computation and the computation of parameter updates in the EGRU scales with the number of events (active units).

3. We demonstrate that the EGRU exhibits task-performance competitive with state-of-the-art recurrent network architectures on real-world machine learning benchmarks.

4. We empirically show that EGRU exhibits high levels of activity-sparsity during both inference (forward pass) and learning (backward pass).

We note here that methods for training with parameter sparsity or improving handling of long-term dependencies are both orthogonal to, and can be combined with our approach (which we plan to do in future work). Our focus, in this paper, is exclusively on using activity-sparsity to increase the efficiency of RNNs, specifically the GRU. We expect our method to be more efficient but not better at handling long-range dependencies compared to the GRU.

The sparsity of the backward-pass overcomes one of the major roadblocks in using large recurrent models, which is having enough computational resources to train them. We demonstrate the task performance and activity sparsity of the model implemented in PyTorch, but this formulation will also allow the model to run efficiently on CPU-based nodes when implemented using appropriate software paradigms. Moreover, an implementation on novel neuromorphic hardware like Davies et al. (2018); Höppner et al. (2017), that is geared towards event-based computation, can make the model orders of magnitude more energy efficient (Ostrau et al., 2022).

## 2 RELATED WORK

Activity sparsity in RNNs has been proposed previously (Neil et al., 2017; 2016; Hunter et al., 2022), but only focusing on achieving it during inference. Conditional computation is a form of activity sparsity used in Fedus et al. (2022) to scale a feedforward transformer architecture to 1 trillion parameters. An asynchronous event-based architecture was recently proposed specifically targeted towards graph neural networks (Schaefer et al., 2022). QRNNs (Bradbury et al., 2017), SRUs (Lei et al., 2018) and IndRNNs (Li et al., 2018) target increasing the parallelism in a recurrent network without directly using activity-sparsity. Unlike Fedus et al. (2022) that used a separate network to decide which sub-networks should be active (Shazeer et al., 2017), our architecture uses a unit-local decision making process for the dynamic activity-sparsity. The cost of computation is lower in our model compared to Neil et al. (2017), and can be implemented to have parallel computation of intermediate updates between events, while also being activity sparse in its output.

Models based on sparse communication (Yan et al., 2022) for scalability have been proposed recently for feedforward networks, using locality sensitivity hashing to dynamically choose downstream units

for communicating activations. This is a dynamic form of parameter-sparsity (Hoefler et al., 2021), which is orthogonal to and complementary with our method for activity-sparsity, and can be combined for additional gains. The use of rectified linear units (ReLU) in recurrent networks as in the IRNN (Le et al., 2015) is closely related, and can lead to sparse activations. But current literature lacks an analysis of such models from the context of efficiency. Moreover, the use of ReLU activation requires careful weight initialization, and it seems to lag behind in task performance in comparisons with other RNN models (Li et al., 2018).

Biologically realistic spiking networks (Maass, 1997) are often implemented using event-based updates and have been scaled to huge sizes (Jordan et al., 2018), albeit without any task-related performance evaluation. Models for deep learning with recurrent spiking networks (Bellec et al., 2018; Salaj et al., 2021) mostly focus on modeling biologically realistic memory and learning mechanisms. Moreover, units in a spiking neural network implement dynamics based on biology and communicate solely through unitary events, while units in an EGRU send real-valued signals to other units, and have different dynamics. A sparse learning rule was recentl y proposed (Bellec et al., 2020) that is a local approximation of backpropagation through time, but not event-based.

The theoretical analysis of the event-based learning rule for the continuous time EGRU is inspired by, and a generalization of the analysis in Wunderlich and Pehle (2021) for spiking neurons. As in that paper, we use the adjoint method for ordinary differential equations (ODEs) Pontryagin et al. (1962); Chen et al. (2018) combined with sensitivity analysis for hybrid discrete/continuous systems (Galán et al., 1999; Chen et al., 2020). Using surrogate gradients for backpropagating through the non-differential threshold function was originally proposed for feedforward spiking networks in Esser et al. (2016) and developed further in Bellec et al. (2018); Zenke and Ganguli (2018). The sparsity of learning with BPTT when using appropriate surrogate gradients in a discrete-time feed-forward spiking neural network was recently described in Perez-Nieves and Goodman (2021).

A continuous time version of sigmoidal RNNs was proposed in Beer (1995) and for GRUs in De Brouwer et al. (2019). The latter used a Bayesian update for network states when input events were received, but the network itself was not event-based. As in Neil et al. (2016); Lechner and Hasani (2020), the focus there was on modeling irregularly spaced input data, and not on event-based network simulation or activity-sparse inference and training. Chang et al. (2019) also recently proposed a continuous time recurrent network for more stable learning, without event-based mechanics. GRUs were formulated in continuous time for analyzing its autonomous dynamics in Jordan et al. (2021).

## 3 EVENT-BASED GRU

### 3.1 TIME-SPARSE GRU FORMULATION

We base our model on the GRU (Cho et al., 2014), illustrated for convenience in Fig. 1A. It consists of internal gating variables for updates ($\mathbf{u}$) and a reset ($\mathbf{r}$), that determine the behavior of the internal state $\mathbf{y}$. The state variable $\mathbf{z}$ determines the interaction between external input $\mathbf{x}$ and the internal state. The dynamics of a layer of GRU units, at time step $t$, is given by the set of vector-valued update equations:

$$\mathbf{u}^{\langle t\rangle}=\sigma\Big(\mathbf{W}_u\Big[\mathbf{x}^{\langle t\rangle},\mathbf{y}^{\langle t-1\rangle}\Big]+\mathbf{b}_u\Big),\quad \mathbf{r}^{\langle t\rangle}=\sigma\Big(\mathbf{W}_r\Big[\mathbf{x}^{\langle t\rangle},\mathbf{y}^{\langle t-1\rangle}\Big]+\mathbf{b}_r\Big),$$

$$\mathbf{z}^{\langle t\rangle}=g\Big(\mathbf{W}_z\Big[\mathbf{x}^{\langle t\rangle},\mathbf{r}^{\langle t\rangle}\odot\mathbf{y}^{\langle t-1\rangle}\Big]+\mathbf{b}_z\Big),\quad \mathbf{y}^{\langle t\rangle}=\mathbf{u}^{\langle t\rangle}\odot\mathbf{z}^{\langle t\rangle}+(1-\mathbf{u}^{\langle t\rangle})\odot\mathbf{y}^{\langle t-1\rangle}, \tag{1}$$

where $\mathbf{W}_{u/r/z}$, $\mathbf{b}_{u/r/z}$ denote network weights and biases, $\odot$ denotes the element-wise (Hadamard) product, and $\sigma\left(\cdot\right)$ is the vectorized sigmoid function. The notation $\big[\mathbf{x}^{\langle t\rangle},\mathbf{y}^{\langle t-1\rangle}\big]$ denotes vector concatenation. The function $g(\cdot)$ is an element-wise nonlinearity, typically the hyperbolic tangent.

We introduce an event generating mechanisms by augmenting the GRU with a rectifier (a thresholding function). See Fig. 1B for an illustration. With this addition the internal state variable $y_i^{\langle t\rangle}$ is nonzero only when the internal dynamics reach a threshold $\vartheta_i$ and is cleared immediately afterwards, thus making $y_i^{\langle t\rangle}$ event-based. Formally, we add an auxiliary internal state $c_i^{\langle t\rangle}$ to the model, and replace $\mathbf{y}^{\langle t\rangle}=(y_1^{\langle t\rangle},y_2^{\langle t\rangle},...)$ with the event-based form

$$y_i^{\langle t\rangle} = c_i^{\langle t\rangle} H\Big(c_i^{\langle t\rangle}-\vartheta_i\Big) \quad\text{with}\quad c_i^{\langle t\rangle} = u_i^{\langle t\rangle}z_i^{\langle t\rangle}+(1-u_i^{\langle t\rangle})c_i^{\langle t-1\rangle}-y_i^{\langle t-1\rangle}, \tag{2}$$

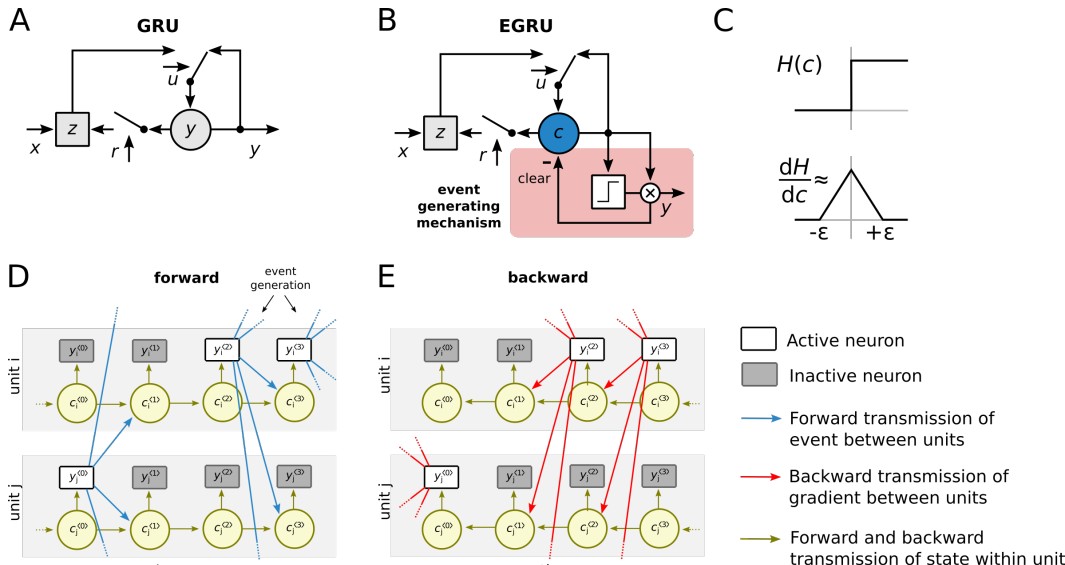

**Figure 1:** Illustration of EGRU. **A:** A single unit of the original GRU model adapted from Cho et al. (2014). **B:** EGRU unit with event generating mechanism. **C:** Heaviside function and surrogate gradient. **D:** Forward state dynamics for two EGRU units ($i$ and $j$). **E:** Activity-sparse backward dynamics for two EGRU units ($i$ and $j$). Note that we only have to backpropagate through units that were active or whose state was close to the threshold at each time step.

where $H(\cdot)$ is the Heaviside step function and $\vartheta_i > 0$ is a trainable threshold parameter. $H(\cdot)$ is the threshold gating mechanism here, generating a single non-zero output when $c_i^{\langle t \rangle}$ crosses the threshold $\vartheta_i$. That is, at all time steps $t$ with $c_i^{\langle t \rangle} < \vartheta_i, \forall i$, we have $y_i^{\langle t \rangle} = 0$. The $-y_i^{\langle t-1 \rangle}$ term in Eq. (2) makes emission of multiple consecutive events by the same unit unlikely, hence favoring overall sparse activity. With this formulation, each unit only needs to be updated when an input is received either externally or from another unit in the network. This is because, if both $x_i^{\langle t \rangle} = y_i^{\langle t-1 \rangle} = 0$ for the $i$-th unit, then $u_i^{\langle t \rangle}, r_i^{\langle t \rangle}, z_i^{\langle t \rangle}$ are essentially constants, and hence the update for $y_i^{\langle t \rangle}$ can be retroactively calculated efficiently on the next incoming event.

### 3.2 SPARSE APPROXIMATE BPTT

The threshold activation function $H(c)$ to decide whether to emit an event in Eq. (2) is not differentiable at the threshold $\vartheta_i$. We define a surrogate gradient at that point for calculating the backpropagated gradients. The surrogate gradient is defined as a piece-wise linear function that is non-zero for values of state $c_i$ between $\vartheta_i + \varepsilon$ and $\vartheta_i - \varepsilon$ as shown in the inset in Fig. 1C. Since the surrogate gradient is zero whenever the internal state of the unit is below $\vartheta_i - \varepsilon$, the backpropagated gradients are also 0 for all such units, making the backward-pass sparse (see Fig. 1D, E for an illustration). Note that the case where the internal unit state is above $\vartheta_i + \varepsilon$ tends to occur less often, since the unit will emit an event and the internal state will be cleared at the next simulation step in that case.

### 3.3 COMPUTATION AND MEMORY REDUCTION DUE TO SPARSITY

For the forward pass of the EGRU, an activity sparsity of $\alpha$ (i.e. an average of $\alpha$ events per simulation step) leads to the reduction of multiply-accumulate operations (MAC), by factor $\alpha$. We focus on MAC operations, since they are by far the most expensive compute operation in these models. MACs are also relevant for our focus on resource constrained systems which are more likely to be based on CPUs rather than GPUs. If optimally implemented an activity sparsity of 80% will require 80% fewer MAC operations compared to a standard GRU. Computation related to external input is only performed at input times, and hence is as sparse as the input, both in time and space. During the backward pass, a similar factor of computational reduction is observed, based on the backward-pass sparsity $\beta$ which is, in general, less than $\alpha$. This is because, when the internal state value is not within $\pm\varepsilon$ of the threshold $\vartheta$, the backward pass is skipped, as described in section 3.2. Since our backward pass is also sparse,

we need to store only $\beta$ fraction of the activations for later use, hence also reducing the memory usage. In all our experiments, we report activity-sparsity values calculated through simulations.

## 4 THEORETICAL ANALYSIS OF THE EGRU

To further analyze the dynamics of EGRU we develop a continuous-time version of the model. This allows us to study the model parameters as a dynamical system.

### 4.1 LIMIT TO CONTINUOUS TIME

Eq. (1) of the discrete time model considers the GRU dynamics only at integer time points, $t_0 = 0$, $t_1 = 1$, $t_2 = 2$,.... However, in general it is possible to express the GRU dynamics for an arbitrary time step $\Delta t$, with $t_n = t_{n-1} + \Delta t$. The discrete time GRU dynamics can be intuitively interpreted as an Euler discretization of an ordinary differential equation (ODE) (Jordan et al., 2021) (see Supplement), which we extend further to formulate the EGRU. This is equivalent to taking the continuous time limit $\Delta t \to 0$ to get dynamics for the internal state $\mathbf{c}(t)$. In the resulting dynamical system equations inputs cause changes to the states only at the event times, whereas the dynamics between events can be expressed through ODEs. To arrive at the continuous time formulation we introduce the neuronal activations $\mathbf{a}_u(t)$, $\mathbf{a}_r(t)$ and $\mathbf{a}_z(t)$, with

$$\mathbf{u}(t) = \sigma(\mathbf{a}_u(t)), \quad \mathbf{r}(t) = \sigma(\mathbf{a}_r(t)), \quad \mathbf{z}(t) = g(\mathbf{a}_z(t)),$$

$$\text{with dynamics} \quad \tau_s \dot{\mathbf{a}}_X = -\mathbf{a}_X - \mathbf{b}_X, \quad X \in \{u, r, z\} \tag{3}$$

$$\text{and} \quad \tau_m \dot{\mathbf{c}}(t) = \mathbf{u}(t) \odot (\mathbf{z}(t) - \mathbf{c}(t)) = F(t, \mathbf{a}_u, \mathbf{a}_r, \mathbf{a}_z, \mathbf{c}), \tag{4}$$

where $\tau_s$ and $\tau_m$ are time constants, $\mathbf{c}(t)$, $\mathbf{u}(t)$ and $\mathbf{z}(t)$ are the continuous time analogues to $\mathbf{c}^{\langle t \rangle}$, $\mathbf{u}^{\langle t \rangle}$ and $\mathbf{z}^{\langle t \rangle}$, and $\dot{\mathbf{a}}_X$ denotes the time derivative of $\mathbf{a}_X$. The boundary conditions are defined for $t = 0$ as $\mathbf{a}_X(0) = \mathbf{c}(0) = \mathbf{0}$. The function $F$ in Eq. (4) determines the behavior of the EGRU between event times, i.e. when $\mathbf{x}(t) = \mathbf{0}$ and $\mathbf{y}(t) = \mathbf{0}$. Nonzero external inputs and internal events cause jumps in $\mathbf{c}(t)$ and $\mathbf{a}_X(t)$. For theoretical tractability, we add a decay term $-\mathbf{a}_X$ to the ODE in Eq. (3), which is implemented with a small or zero time constant in the discrete time model.

To describe these dynamics we introduce the set of internal events $\mathbf{e}$, $e_k \in \mathbf{e}$, $e_k = (s_k, n_k)$, where $s_k$ are the continuous (real-valued) event times, and $n_k$ denotes which unit got activated. The formulation of the event generating mechanisms Eq. (2) introduced above can be expressed as event $e_k$ that is triggered whenever $c_{n_k}(t)$ reaches $\vartheta$. More precisely:

$$(s_k, n_k) : c_{n_k}^-(s_k) = \vartheta_{n_k}, \tag{5}$$

where the superscript $.^-$ $(.^+)$ denotes the quantity just before (after) the event. The clearing mechanism in continuous time is expressed as resetting $c_i(s)$ to zero right after event times $s$. This is because in continuous time the exact time $s$ at which the internal variable $c_i(s)$ reaches the threshold $(c_i(s) = \vartheta_i)$ can be determined with very high precision. Therefore, the value of $c_i(s)$ and the instantaneous amplitude of $y_i(s)$ simultaneously approach $\vartheta_i$ at time point $s$, so that the $-y_i$ term in Eq. (2) effectively resets $c_i(s)$ to zero, right after an event was triggered.

At the time of this event, the activations of all the units $m \neq n_k$ connected to unit $n_k$ experiences a jump in its state value. The jump for $a_{X,m}$ is given by:

$$a_{X,m}^+(s_k) = a_{X,m}^-(s_k) + w_{X,mn_k} r_{X,n_k} c_{n_k}^-(s_k), \tag{6}$$

where $X \in \{u, r, z\}$, $\mathbf{r}_X = 1$ when $X \in \{u, z\}$ and $\mathbf{r}_X = \mathbf{r}$ when $X = \{r\}$. This is equivalent to $y_i = c_{n_k}^-$ being the output of each network unit. A similar jump is experienced on arrival of an external input, using the appropriate input weights instead (see Supplement for specifics). The event-based asynchronous nature of this update scheme can be formalized in the following proposition.

**Proposition 1.** *Let the dynamics of $\mathbf{u}(t)$ and $\mathbf{c}(t)$ be given by (3) and (4). The unperturbed dynamic (in the absence of input) of internal states $\mathbf{c}(t)$ are decoupled (the variables of $\mathbf{c}(t)$ do not interact with each other).*

*Proof.* The proof follows directly from the definition (4). While variables of single units interact (e.g. $u_i(t)$ and $c_i(t)$), the dynamics of variables of all units are decoupled from each other in the absence of inputs. □

## 4.2 EVENT-BASED GRADIENT-DESCENT USING ADJOINT METHOD

To show that the EGRU gradient updates are event-based, we define the loss over duration $T$ as $\int_0^T \ell_c(\mathbf{c}(t),t)dt$, where $\ell_c(\mathbf{c}(t),t)$ is the instantaneous loss at time $t$. $T$ is a task-specific time duration within which the training samples are given to the network as events, and the outputs are read out. In general $\ell_c(\mathbf{c}(t),t)$ may depend arbitrarily on $\mathbf{c}(t)$, however in practice we choose the instantaneous loss to depend on the EGRU states only at specific output times to adhere to our fully event-based algorithm.

The loss is augmented with the terms containing the Lagrange multipliers $\boldsymbol{\lambda}_c, \boldsymbol{\lambda}_{a_x}$ to add constraints defining the dynamics of the system from Eqs. (3), (4). The total loss $\mathcal{L}$ thus reads

$$\mathcal{L} = \int_0^T \left[ \ell_c(\mathbf{c}(t),t) + \boldsymbol{\lambda}_c \cdot (\tau_m \dot{\mathbf{c}}(t) - F(t,\mathbf{a}_u,\mathbf{a}_r,\mathbf{a}_z,\mathbf{c})) + \sum_{\mathbf{x}\in\{u,r,z\}} \boldsymbol{\lambda}_{a_x} \cdot (\tau_s \dot{\mathbf{a}}_{\mathbf{x}} + \mathbf{a}_{\mathbf{x}}) \right] dt. \tag{7}$$

The Lagrange multipliers are referred to as the adjoint variables in this context, and may be chosen freely since both $\tau_m \dot{\mathbf{c}}(t) - F(t,\mathbf{a}_u,\mathbf{a}_r,\mathbf{a}_x,\mathbf{c})$ and $\tau_s \dot{\mathbf{a}}_{\mathbf{x}} + \mathbf{a}_{\mathbf{x}}$ are everywhere zero by construction.

We can choose dynamics and jumps at events for the adjoint variables in such a way that they can be used to calculate the gradient $\frac{d\mathcal{L}}{dw_{ji}}$. Calculating the partial derivatives taking into account the discontinuous jumps at event times depends on the local application of the implicit function theorem, which requires event times to be a differentiable function of the parameters. See the Supplement for a full derivation.

The time dynamics of the adjoint variables is given by the following equations with a boundary condition of $\boldsymbol{\lambda}_c(T) = \boldsymbol{\lambda}_{a_x}(T) = 0$:

$$\left(\frac{\partial F}{\partial \mathbf{c}}\right)^T \boldsymbol{\lambda}_c - \tau_m \dot{\boldsymbol{\lambda}}_c = 0, \qquad \boldsymbol{\lambda}_{a_x} + \left(\frac{\partial F}{\partial \mathbf{a}_{\mathbf{x}}}\right)^T \boldsymbol{\lambda}_c - \tau_s \dot{\boldsymbol{\lambda}}_{a_x} = 0, \tag{8}$$

for $\mathbf{x} \in \{u,r,z\}$, and $M^T$ denoting the transpose of the matrix $M$. The event updates for the adjoints are described in the Supplement. In practice, the integration of $\boldsymbol{\lambda}$ is done backwards in time. In analogy to Proposition 1 we find the following property of the forward dynamics.

**Proposition 2.** *Let $\boldsymbol{\lambda}_c(t)$ and $\boldsymbol{\lambda}_{a_x}(t)$ be given by the adjoint dynamics Eq. (8). Let the initial conditions of $\boldsymbol{\lambda}_c$ be given by $\boldsymbol{\lambda}_c(s_k)$. Then the system of differential equations $\boldsymbol{\lambda}_c(t)$ for $s_{k-1} < t < s_k$ is decoupled (the variables of $\boldsymbol{\lambda}_c$ do not interact with each other).*

*Proof.* The proof follows from (8). Since the dynamics of $\mathbf{a}_{\mathbf{x}}$ and $\mathbf{c}$ are decoupled (unit-wise) between events as shown in Proposition 1, the dynamics of $\boldsymbol{\lambda}_c$ are also decoupled. The term $(\partial F/\partial \mathbf{a}_{\mathbf{x}})^T \boldsymbol{\lambda}_c$ reduces to a vector without cross-unit interactions as shown in the Supplement. $\qquad\square$

For the recurrent weights $w_{\mathbf{x},ij}$ from the different parameter matrices $W_{\mathbf{x}}$ for $\mathbf{x} \in u,r,z$, we can write the weight updates using only quantities calculated at events $e_k$ as $\Delta w_{\mathbf{x},ij} = \frac{\partial}{\partial w_{\mathbf{x},ij}} \mathcal{L}(\mathbf{W})$. We find the following property of the weight updates.

**Theorem 1.** *Let $\boldsymbol{\lambda}_c(t)$ and $\boldsymbol{\lambda}_{a_x}(t)$ be given by the adjoint dynamics Eq. (8) such that Propositions 1 and 2 hold. Let $\boldsymbol{\mu}_X^-(s_k) = \mathbf{r}_{\mathbf{x}}^-(s_k) \odot \mathbf{c}^-(s_k)$. Then the weight updates $\Delta w_{\mathbf{x},ij} = \frac{\partial}{\partial w_{\mathbf{x},ij}} \mathcal{L}(\mathbf{W})$ are independent of each other and fully determined by $\boldsymbol{\mu}_X^-(s_k)$ and $\boldsymbol{\lambda}_{a_x}^-(s_k)$, the set of EGRU states evaluated at event times $s_k$, i.e. $\Delta w_{\mathbf{x},ij} = -\tau_s \sum_{k=1}^K \boldsymbol{\mu}_X^-(s_k) \otimes \boldsymbol{\lambda}_{a_x}^+(s_k)$.*

*Proof sketch.* Here, $\otimes$ is the outer product, $\mathbf{c}^-$ refers to the value of $\mathbf{c}(t)$ just before event $e_k$, $\mathbf{r}_{\mathbf{x}}^- = 0$ for $\mathbf{x} \in \{u,z\}$ and equal to the value of $\mathbf{r}(t)$ just before event $e_k$ for $\mathbf{x} = \{r\}$, $\boldsymbol{\lambda}_{a_x}^+$ refers to the value of the adjoint variable $\boldsymbol{\lambda}_{a_x}(t)$ just after the event $e_k$, and $K = |\mathbf{e}|$ is the total number of events. Thus, the values of $\mathbf{r}(t), \mathbf{c}(t)$ need to be stored only at event times, and $\boldsymbol{\lambda}_{a_x}(t)$ needs to be calculated only at these times, making the gradient updates event-based. See the Supplement for more details to the proof and for update rules for the input weights and biases. Finally we find the following corollary for the algorithmic complexity of the learning algorithm.

**Corollary 1.** *Given $\boldsymbol{\mu}_X^-(s_k)$ and $\boldsymbol{\lambda}_{a_x}^-(s_k)$, the time complexity of parameter update computation $\Delta w_{\mathbf{x},ij}$ grows linearly with $K$ (the number of events).*

*Proof.* The proof follows from Theorem 1. The computation required for the outer product $\boldsymbol{\mu}_X^-(s_k) \otimes \boldsymbol{\lambda}_{a_x}^+(s_k)$ depends on the network size and is independent of the total number of events $K$. The sum thus grows linearly with $K$. $\qquad\square$

## 5 RESULTS

### 5.1 GESTURE PREDICTION

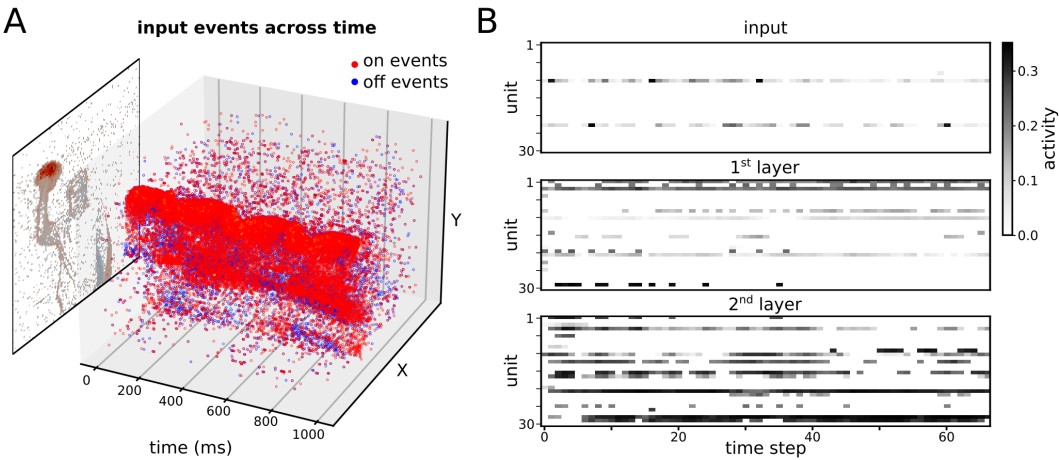

**Figure 2: A:** Illustration of DVS gesture classification data for an example class (right hand wave). On (red) and off (blue) events are shown over time and merged into a summary image for illustration (not presented to the network). **B:** Sparse activity of input and EGRU units (random subset of 30 units shown for each layer).

We evaluated our model on gesture prediction, which is a popular real-world benchmark for RNNs and widely used in neuromorphic research. The DVS128 Gesture Dataset (Amir et al., 2017), provides sparse event-based inputs which enables us to demonstrate our model's performance and computational efficiency. The dataset contains 11 gestures from 29 subjects recorded with a DVS128 event camera (Lichtsteiner et al., 2008). Each event encodes a relative change of illumination and is given as spatio-temporal coordinates of X/Y position on the $128 \times 128$-pixel sensor and time stamp. We combined the raw event times into 'frames' by binning them over time windows of 25 ms, and then downscaled them to $32 \times 32$ pixels using a maxpool layer.

| model | hidden dim | para- meters | effective MAC | test accuracy | activity sparsity | backward sparsity |
|---|---|---|---|---|---|---|
| LSTM (He et al., 2020) | 512 | 7.4 M | 3.9 M | 86.8% | - | - |
| AlexNet+LSTM+DA (Innocenti et al., 2021) | 256 | 8.3 M | 601.3 M | 97.7% | - | - |
| GRU | 1024 | 15.7 M | 10.6 M | 88.1% | 0% | - |
| EGRU | 512 | 5.5 M | **0.98 M** | 88.0% | 83.8% | 46.8% |
| GRU+DA | 1024 | 15.7 M | 10.6 M | 95.1% | 0% | - |
| EGRU+DA | 1024 | 15.7 M | 3.2 M | 97.1% | 78.8% | 58.2% |
| CNN+GRU+DA | 136 | 2.1 M | 79.1 M | 97.4% | 0% | - |
| CNN+EGRU+DA | 795 | 4.8 M | 80.6 M | **97.8%** | 76.4% | 72.3% |

**Table 1:** Model comparison for the DVS Gesture recognition task. Effective number of MAC operations as described in section 3.3. DA stands for Data Augmentation.

Comparison of model performance on gesture prediction is presented in Table 1. The backward sparsity as described in Section 3.3 was calculated at epoch 100. EGRU consistently outperformed GRU networks of the same size on this task by a small margin. Adding data augmentation (DA) by applying random crop, translation, and rotation, as previously done in Innocenti et al. (2021), increased the performance of pure RNN EGRU architecture to over 97%, coming close to state-of-the-art architectures without costly AlexNet pre-processing. We improved this result with a CNN feature extraction head adapted from ALexNet, DropConnect (Wan et al., 2013) applied to the hidden-to-hidden weights and Zone-out (Krueger et al., 2017), outperforming Innocenti et al. (2021) with far fewer parameters and an order of magnitude reduction in MAC operations. Further experimental details, ablation studies and statistics over different runs can be found in the supplement sections E.1, E.1.1 and tables S1, S4 respectively.

## 5.2 Sequential MNIST

We evaluated the EGRU on the sequential MNIST and permuted sequential MNIST tasks (Le et al., 2015), which are widely used benchmarks for RNNs. In the sequential MNIST task, the MNIST handwritten digits were given as input one pixel at a time, and at the end of the input sequence, the network output was used to classify the digit. For the permuted sequential MNIST task, the pixels are permuted using a fixed permutation before being given as input in the same way. We trained a 1-layer EGRU with 590 units (matching the number of parameters with a 512 unit LSTM). We did not use any regularisation to increase sparsity in this task, so that we could test how much sparsity, both forward and backward, arises naturally in the EGRU.

In Table 2, we report the results of discrete-time EGRU along with other state-of-the-art architectures. EGRU achieved a task performance comparable to previous architectures while using much fewer operations (more than an 5-fold reduction in effective MAC operations compared to GRU). Further experimental details, and statistics over different runs can be found in the supplement sections E.2 and table S5 respectively.

| dataset | model | hidden dim | para- meters | effective MAC | test accuracy | activity sparsity | backward sparsity |
|---|---|---|---|---|---|---|---|
| sMNIST | coRNN (Gu et al., 2020) | 256 | 134 K | 262 K | **99.4%** | - | - |
|  |  | 512 | 1 M | 1 M | 98.9% | - | - |
|  | GRU | 590 | 1 M | 1 M | 98.8% | - | - |
|  | EGRU | 590 | 1 M | **226 K** | 98.3% | 72.1% | 27.4% |
| psMNIST | coRNN (Gu et al., 2020) | 256 | 134 K | 262 K | **97.3%** | - | - |
|  |  | 512 | 1 M | 1 M | 95.1% | - | - |
|  | EGRU | 590 | 1 M | **195 K** | 95.1% | 82.0% | 8.4% |

**Table 2:** Model comparison on sequential MNIST (sMNIST) and permuted sequential MNIST (psMNIST) task. Top-1 test scores, given as percentage accuracy, where higher is better. coRNN is the model described in Rusch and Mishra (2021).

## 5.3 Language Modeling

We evaluated our model on language modeling tasks based on the PennTreebank (Marcus et al., 1993) dataset and the WikiText-2 dataset (Merity et al., 2017). We focused exclusively on the RNN model in this work, and did not consider techniques such as neural cache models (Grave et al., 2017), Mixture-of-Softmaxes (Yang et al., 2018) or dynamic evaluation (Krause et al., 2018), all of which can be used on top of our model. A strong baseline for gate-based RNN architectures was established by Merity et al. (2018). Similarly, our models consisted of three stacked EGRU cells without skip connections. DropConnect (Wan et al., 2013) was applied to the hidden-to-hidden weights. The weights of the final softmax layer were tied to the embedding layer (Inan et al., 2017; Press and Wolf, 2017). All experimental details, and statistics over different runs can be found in the Supplement in sections E.3.1 and Table S6 respectively.

The results presented in Table 3 show that EGRU achieved performance competitive with AWD-LSTM (Merity et al., 2018). At the same time, EGRU inherently exhibited activity sparsity that reduced the required computational operations (calculated analytically). Another desirable property of EGRU for resource constrained systems is a natural approach to model compression through pruning of least active units. We discuss our experimental findings for this pruning technique using language modeling as a case study in the Supplement (see Sec. E.4).

## 6 Discussion

This work introduces the EGRU, a new form of a recurrent neural network that uses a biologically inspired activity sparsity mechanism. The EGRU extends the GRU with an event-generating mechanism using a threshold and uses surrogate gradients to train the model with BPTT, which leads to computationally sparse forward and backward passes. We theoretically proved that in the continuous time limit, the model's inference and training scales proportionally to the number of events generated and the number of neurons, but not with the number parameters as in conventional RNN models.

| dataset | model | hidden dim | para- meters | effective MAC | test | activity sparsity | backward sparsity |
|---|---|---|---|---|---|---|---|
| PTB | Merity et al. (2018) | 1150 | 24 M | 24.0 M | 57.3 | - | - |
| | Li et al. (2018) | - | 52 M | 51.6 M | **55.2** | - | - |
| | GRU | 1350 | 28 M | 27.6 M | 66.3 | - | - |
| | EGRU | 1350 | 33 M | **9.9 M** | 57.2 | 79.7% | 46.3% |
| | EGRU | 2000 | 55 M | 12.7 M | 57.0 | 84.8% | 42.9% |
| WikiText-2 | Melis et al. (2018) | - | 24 M | - | 65.9 | - | - |
| | Merity et al. (2018) | 1150 | 33 M | 32.0 M | **65.8** | - | - |
| | GRU | 1350 | 39 M | 38.8 M | 71.8 | - | - |
| | EGRU | 1350 | 51 M | **17.5 M** | 70.6 | 76.8 % | 44.2 % |
| | EGRU | 2000 | 74 M | 20.1 M | 68.9 | 82.7 % | 42.2 % |

**Table 3:** Model comparison on PennTreebank and WikiText-2. Test scores are given as perplexities, where lower is better. Effective MAC operations are given for a single time step and consider the layer-wise sparsity in the forward pass. Activity sparsity is given for the trained model to resemble inference sparsity. Backward sparsity is averaged over the whole training. Model parameters were optimized on Penn Treebank and transfered to WikiText-2.

The EGRU achieved competitive task performance on tasks such as gesture recognition, sequential image classification and language modeling while achieving activity sparsity of up to 85% (15% of the units active on average). Scaling up networks for language modeling has shown some of the most promising results in the last few years (Brown et al., 2020), hence, our choice of this task, albeit on a smaller scale. Considering the need for extensive hyperparameter search (Melis et al., 2018) for language modeling, our model achieved promising results while maintaining a high degree of activity-sparsity. For example, our EGRU with 1350 hidden units reached perplexities better than reported by LSTM and GRU baselines, while maintaining the high level of activity-sparsity. To the best of our knowledge, this is the first demonstration of such activity sparsity mechanisms that yields strong benchmark performance compared to baselines.

While we base our model on the GRU due to its simplicity, this formulation can easily be extended to any arbitrary network dynamics, including the LSTM, allowing specialized architectures for different domains. The adjoint method for hybrid systems that we use for analysis here can also be used as a powerful general-purpose tool to formulate cotinuous time models that are activity-sparse and trained using event-based gradient descent updates for any recurrent neural network architectures. Another novel outcome of this paper is that this theory can handle inputs in continuous time as events, which is very intuitive, hence providing an alternative to the more complex controlled differential equations (Kidger et al., 2020). The EGRU can also be used for irregularly spaced sequential data quite naturally.

The compute efficiency of this model can directly translate into gains in energy efficiency when implemented using event-based software primitives. The model will work well on heterogeneous compute resources, including pure CPU nodes, neuromorphic devices such as Intel's Loihi (Davies et al., 2018) and SpiNNaker 2 (Höppner et al., 2017), that can achieve orders of magnitude higher energy efficiency, as well as on deep learning hardware that support dynamic sparsity, such as the Graphcore system (Jia et al., 2019). On neuromorphic devices with on-chip memory in the form of a crossbar array, the activity sparsity directly translates into energy efficiency. For larger models that need off-chip memory, activity-sparsity needs to be combined with parameter-sparsity to reduce energy-intensive memory access operations.

In summary, with the motivation of building scalable, energy-efficient deep recurrent models, we presented an activity sparsity mechanism that reduces the required compute for both inference and learning. We demonstrated that the EGRU – our GRU-based model using this activity sparsity mechanism – is a competitive alternative to GRU and LSTM models, especially for resource constrained systems and neuromorphic devices. In future work, we plan to combine this activity sparsity mechanism with methods for parameter sparsity and learning long-range dependencies and implement them on neuromorphic hardware to realise the efficiency gains and scalability of recurrent neural network architectures.

## 7 REPRODUCIBILITY

We ensure that the results presented in this paper are easily reproducible using just the information provided in the main text as well as the appendix. Details of the models used in our experiments are presented in the main paper and further elaborated in the appendix. We provide additional experimental details, ablation studies, and statistics over multiple runs in the appendix Section E. We use publicly available libraries (Appendix section G) and Datasets (Appendix section F) in our experiments. We will further provide the source code to the reviewers and ACs in an anonymous repository once the discussion forums are opened. The included code will also contains "readme" texts to facilitate easy reproducibility. The theoretical analysis provided in Section 4 is derived in the appendix along with the event-based learning rule.

## 8 ACKNOWLEDGEMENTS

The authors gratefully acknowledge the GWK support for funding this project by providing computing time through the Center for Information Services and HPC (ZIH) at TU Dresden. We acknowledge the use of Fenix Infrastructure resources, which are partially funded from the European Union's Horizon 2020 research and innovation programme through the ICEI project under the grant agreement No. 800858. AS is funded by the Ministry of Culture and Science of the State of North Rhine-Westphalia, Germany. KKN is funded by the German Federal Ministry of Education and Research (BMBF) within the KI-ASIC project (16ES0996). MS is fully funded by a grant of the Bosch Research Foundation. CM receives fundeding from the German Research Foundation (DFG, Deutsche Forschungsgemeinschaft) as part of Germany's Excellence Strategy – EXC 2050/1 – Project ID 390696704 – Cluster of Excellence "Centre for Tactile Internet with Human-in-the-Loop" (CeTI) of Technische Universität Dresden. DK is funded by the German Federal Ministry of Education and Research (BMBF) within the project EVENTS (16ME0733). The authors would like to thank Darjan Salaj, Melika Payvand, Markus Murschitz, Franz Scherr, Robin Schiewer for helpful comments on this manuscript. AS would also like to thank Darjan Salaj and Franz Scherr for insightful early discussions, and Laurenz Wiskott for institutional support.

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
