# OpenReview forum: "Efficient recurrent architectures through activity sparsity and sparse back-propagation through time"
_ICLR.cc/2023/Conference — ICLR 2023 notable top 25%_

### Official Review · Reviewer_KSan · 2022-10-24

**Confidence:** 4
**Correctness:** 4
**Technical Novelty And Significance:** 3
**Empirical Novelty And Significance:** 3
**Recommendation:** 8

**Clarity, Quality, Novelty And Reproducibility:**

The writing is quite clear and the ideas and motivations are well-presented.
Although the idea is not completely novel, I feel this is a valuable piece of work as it presents an architecture leveraging that idea. As mentioned above, I would be keen to see an connection/comparison with the IRNN architecture.
The work seems to be reproducible, although I haven't read the experiment details in the appendix. Also, the authors propose to make the code base available which should be a valuable resource for the community.

**Strength And Weaknesses:**

Strengths:
1. The paper presents a strong motivations for their solution and is overall well-written.
2. The theoretical results present strong evidence of why their solution enables efficient scaling up of models.
3. The EGRU architecture contributes to reducing the memory footprint of BPTT, which is indeed a significant step towards making BPTT implementable in long sequences (instead of relying of truncated BPTT). It would be interesting to see how EGRU performs with some of the biologically-proposed temporal credit assignment rules, that rely on erasing out credit signals carried over long temporal horizons.

Weaknesses:
1. I feel the EGRU is quite similar to the Identity-RNNs proposed by Le, Jaitly & Hinton in 2015, wherein they used a ReLU non-linearity with RNNs (already cited by the author in the context of sequential MNIST experiments). Although EGRU has a mechanism to avoid multiple firing of the same unit, which is different from the Identity-RNN formulation, I would be curious to know any corresponding links/ comparison to the IRNN architecture.
2. Since one of the central claims that the authors make relates to scalability, it would be nice to see how the performance and activity/backward sparsitiy changes when the number of hidden units in EGRU are changed. A trend to empiricially demonstrate how these quantities change with scaling up the network would be useful.
3. A minor point (feel free to ignore if it is more along the directions in future work): it would be interesting to see how the sparsity changes across training. The authors currently report the average sparsity of the network but I would be curious to know if the operational sparsity is lower than that of an untrained network because the network eventually converged to such a solution.

**Summary Of The Paper:**

The authors present an event-based version of GRUs that allows for sparse forward and backward passes that achieves similar (sometimes even better) task performance than vanilla GRU but at a lower compute cost. The authors also present theoretical results indicating that the computation complexity doesn't scale with the network size, but instead with the number of events. Although it is unclear how does the number of events scale with network size (I personally feel that it would depend on the task and training loss function, and is therefore hard to make a generic statement), the implications of this result are significant in terms of efficiently scaling up network size for solving complicated tasks. In this work, the authors support their claims with empirical evidence on gesture recognition, sequential MNIST and language modeling tasks.

**Summary Of The Review:**

Overall, I would recomment accepting the paper. I feel the paper can be slightly improved by updating the motivation by talking about connections to ReLU non-linearity for RNNs, which are also supposed to sparsify the activations. I have currently rated the paper to be marginally above the acceptance threshold, but if the authors can address some of the weaknesses mentioned above, I am happy to increase it to 8.

---

> ### Author Response · Authors · 2022-11-16
> **Response to reviewer KSan**
>
> We thank the reviewer for their constructive and insightful feedback and the overall positive evaluation of the paper. Responding to their feedback:
>
> > I feel the EGRU is quite similar to the Identity-RNNs proposed by Le, Jaitly & Hinton in 2015, wherein they used a ReLU non-linearity with RNNs (already cited by the author in the context of sequential MNIST experiments). Although EGRU has a mechanism to avoid multiple firing of the same unit, which is different from the Identity-RNN formulation, I would be curious to know any corresponding links/ comparison to the IRNN architecture.
>
> Thank you for pointing out this very interesting relation between the IRNN and EGRU. Apart from the different unit dynamics and lack of the clear mechanism (as the reviewer points out) we agree that the IRNN can potentially have very similar properties as the EGRU — although this doesn’t seem to have been analysed in literature so far to our knowledge.
>
> But the IRNN also seems to compare unfavourable with other recurrent architectures in more recent comparisons — for example, in [1].
>
> That being said, this is definitely an interesting connection and we have added this discussion to the related work.
>
> [1] S. Li, W. Li, C. Cook, and Y. Gao. Deep Independently Recurrent Neural Network (IndRNN). arXiv:1910.06251 [cs], Dec. 2020. URL [http://arxiv.org/abs/1910.06251](http://arxiv.org/abs/1910.06251).
>
> > Since one of the central claims that the authors make relates to scalability, it would be nice to see how the performance and activity/backward sparsitiy changes when the number of hidden units in EGRU are changed. A trend to empiricially demonstrate how these quantities change with scaling up the network would be useful.
>
> > Although it is unclear how does the number of events scale with network size (I personally feel that it would depend on the task and training loss function, and is therefore hard to make a generic statement)
>
> Supplementary figure S1 shows an initial analysis on how the number of active units per timestep varies with the size of the network. We are running more comprehensive experiments and plan to have further results in the final version of the paper.
>
> > A minor point (feel free to ignore if it is more along the directions in future work): it would be interesting to see how the sparsity changes across training. The authors currently report the average sparsity of the network but I would be curious to know if the operational sparsity is lower than that of an untrained network because the network eventually converged to such a solution.
>
> Supplementary figure S4 shows both the forward and backward sparsity across training for each of the layers in the PTB language modelling task.

---

> > ### Comment · Reviewer_KSan · 2022-12-05
> > **Response to authors**
> >
> > I would like to thank the authors for their response.
> > Given the updates to the manuscript and their response, I have increased my score.

---

### Official Review · Reviewer_MdUd · 2022-10-24

**Confidence:** 4
**Correctness:** 3
**Technical Novelty And Significance:** 3
**Empirical Novelty And Significance:** 3
**Recommendation:** 8

**Clarity, Quality, Novelty And Reproducibility:**

The article is generally very clearly written and the model is well presented and theoretically analyzed. Minor: one redundant sentence in the first sentences of paragraphs 4 and 4. 1. In addition, the same number of digits should be used for precision in Table 1.

**Strength And Weaknesses:**

The strength of the paper lies in the focus on the sparsity inducing mechanism and its theoretical and numerical application. In particular, the success of several benchmarks such as DVS gesture, sequential MNIST and natural language processing clearly demonstrate the advantage of this method.
One weakness of the paper lies in the lack of analysis of the encoding resulting from the introduction of the sparsity induction mechanism. For example, Figure 2 shows the activity produced by a segment of the DVS gesture dataset, the output of which in the different layers is qualitatively very different from the raster plots observed in biology. Further analysis of the average frequency of completion could help to understand how this could be improved, for example by incorporating homeostatic mechanisms that optimize the information carried by each neuron (for simplicity, a neuron that does not respond or always responds carries no information about the sensory input). Also, what is the influence of thresholding on accuracy? This would be an added value to the article.

**Summary Of The Paper:**

Authors show that by including the biologically grounded constraint that neural activity is sparse, learning in a recurrent neutral network can be improved, both in the areas of computation (memory footprint,...) and accuracy. Using a continuous time extension of the recurrent neural network model, they theoretically prove the properties of the proposed model, and in particular the modulation of the computational cost with sparsity of the activity. The numerical results provide encouraging results for applying this methodology to generic networks and in particular their use in neuromorphic hardware.


**Summary Of The Review:**

In summary, the authors provide an important and novel contribution to recurrent neural networks that have important applications to neuromorphic hardware. The work could be further improved by showing mechanically why their solution is improved over SOTA.

---

> ### Author Response · Authors · 2022-11-16
> **Response to reviewer MdUd**
>
> We thank the reviewer for their constructive and insightful feedback and the overall positive evaluation of the paper. Responding to their feedback:
>
> > One weakness of the paper lies in the lack of analysis of the encoding resulting from the introduction of the sparsity induction mechanism. For example, Figure 2 shows the activity produced by a segment of the DVS gesture dataset, the output of which in the different layers is qualitatively very different from the raster plots observed in biology.
>
> While the discrete event-based unit communication is biologically inspired in construction, we haven’t made an attempt to stay close to the biological visual system with the overall architecture — for example we use only recurrent layers for processing the DVS input in the visualised example.  We plan to compare our model activity to biological data to understand the differences in more detail in future work.
>
> > Further analysis of the average frequency of completion could help to understand how this could be improved, for example by incorporating homeostatic mechanisms that optimize the information carried by each neuron (for simplicity, a neuron that does not respond or always responds carries no information about the sensory input).
>
> The use of homeostatic mechanisms to optimise communication is a very interesting suggestion, and we thank the reviewer for bringing this up. We have added analysis of neuron activity in the supplement (Fig S3) and it looks like the network could benefit from such an optimisation. We will consider this for future work.
>
> > Also, what is the influence of thresholding on accuracy?
>
> We are currently running comprehensive experiments for studying the role of initialisation of the thresholds and will add the results to the final version of the paper.
>
> Our observation during the experiments has been that initialising with mean around zero works better for LM, while a uniform initialisation works better for DVS.
>
> > Minor: one redundant sentence in the first sentences of paragraphs 4 and 4. 1. In addition, the same number of digits should be used for precision in Table 1.
>
> We have fixed these issues in the updated version of the paper.
>
> > The work could be further improved by showing mechanically why their solution is improved over SOTA.
>
> Thank you for the suggestion. We plan to analyse why our model works better for these tasks in further detail in future work. Since the DVS Gesture recognition dataset is also event-based we think that there might be a synergy between our model and the data which could be responsible for the high task performance we achieve. Activity sparsity in our model might also serve as an implicit regulariser, helping with the high test performance.

---

> > ### Author Response · Authors · 2022-11-22
> > **Commenting on model sensitivity to thresholds**
> >
> > We thank the reviewer again for the question
> >
> > >what is the influence of thresholding on accuracy?
> >
> > We ran experiments with different initialization strategies on Penn Treebank. These experiments show the sensitivity of our model to the hyperparameters of our initialization scheme for this language modeling task. To initialize thresholds, we sample $\tau_i \sim \mathcal{N} (\mu, \sigma\sqrt{2})$ and set the threshold value to $\varphi_i = 1 / (1 + \exp(-\tau_i))$ (see supplemental material).
> >
> > Note that $\sigma = 0$ is equivalent to initializing all thresholds to the same value, while choosing $\mu = 0, \sigma = 1$ almost yields a uniform distribution in the interval $[0, 1]$.
> >
> > Our experiments show that language modeling benefits from initializing thresholds near zero ($\mu = -4$). Trainable thresholds slightly outperform non-trainable thresholds. The gap depends on the initialization and is fairly small for the best initialization strategies. This is not very much surprising as the model is able to counteract constant thresholds with bias terms in the GRU equations.
> >
> > We will include these experiments (mean and std over 3 runs per hyperparameter setup) as well as a visualization of the resulting thresholds in the final version of the paper.

---

### Official Review · Reviewer_SZU5 · 2022-10-25

**Confidence:** 4
**Correctness:** 3
**Technical Novelty And Significance:** 3
**Empirical Novelty And Significance:** 3
**Recommendation:** 8

**Clarity, Quality, Novelty And Reproducibility:**

The proposed work is novel and technically sound. The authors have done a good job explaining clearly their proposed EGRU model and presented ample experimental evidence to support the high performance of EGRU under high sparsity and low computational requirements.

**Strength And Weaknesses:**

Strengths:
1) While recurrent neural networks come with several advantages such as low parameters (owing to weight tying), ability to effectively integrate long-range dependencies and model sophisticated functions with nonlinear dynamics, their relevance for applications is restricted by their high computational budget especially during training caused by BPTT. Proposed work addresses this central issue of RNNs by taking inspiration from sparsity in biological neurons.
2) Evaluations performed on DVS gesture recognition, sequential MNIST and language modeling highlight that EGRU can match/outperform related RNN models in task performance while using a relatively much lesser computational requirements.
3) The potential to use neuromorphic hardware to maximize the gains produced by sparse activations of EGRU is very exciting.

Weaknesses:
1) Some of the compared baselines aren't strong enough. E.g. in Table.1, it looks like DA produces a big gain in performance on EGRU. Why did the authors not evaluate GRU or LSTM with DA? What is the CNN+EGRU adding to the comparison? It looks like the effective MACs are much higher on CNN+EGRU networks in return for a marginal gain in test accuracy. On a related note in Table 3, I don't see any difference in number of parameters between EGRU and GRU with the same hidden dim in Table 1. Why is there a difference between matched GRU and EGRU model number of parameters in table 3?
2) What are some of the limitations of the proposed EGRU model? It would be great if the authors could please further highlight room for improvement of EGRU apart from combining with neuromorphic devices in the discussion.

**Summary Of The Paper:**

The authors propose a biologically inspired event-generating mechanism that they integrate within GRU units for enhancing activity sparsity and reducing train/test compute budget. They show that the time complexity for computing weight updates in the continuous time limit is proportional to the number of events generated in EGRU. The authors show competitive performance of their sparse EGRU network on multiple machine learning benchmarks related to RNNs despite the network producing high activity sparsity and low effective MACs (highly desirable).

**Summary Of The Review:**

This work proposes an effective solution to training RNNs with less computational requirements using activity sparsity found in biological neurons. The authors have theoretically proven that the time complexity to calculate EGRU's weight updates grows linearly with number of events, and evaluated their EGRU model rigorously on a variety of RNN-relevant benchmarks and compared performance against a set of well chosen baselines. It is clear that the proposed EGRU model is capable of achieving high performance with high activity sparsity and potential gains are even larger when EGRU is implemented on suitable neuromorphic hardware. This work is highly promising and is of great relevance to ICLR audience and hence, I recommend acceptance of this paper.

---

> ### Author Response · Authors · 2022-11-16
> **Response to reviewer SZU5**
>
> We thank the reviewer for their constructive comments, and the overall positive evaluation of the paper. Addressing their questions:
>
> > Some of the compared baselines aren't strong enough. E.g. in Table.1, it looks like DA produces a big gain in performance on EGRU. Why did the authors not evaluate GRU or LSTM with DA? What is the CNN+EGRU adding to the comparison? It looks like the effective MACs are much higher on CNN+EGRU networks in return for a marginal gain in test accuracy.
>
> We agree that evaluating the suggested combination can make the comparison more comprehensive and have added a comparison with GRU+DA and CNN+GRU+DA to Table 1.
>
> Our goal was to show that with the CNN+EGRU+DA, we were able to achieve state of the art task performance for this task, as well as demonstrate that the EGRU can be used with standard layers such as the CNN. We agree that the cost of adding the CNN may not compensate for the small gain in performance and will highlight this in the final version.
>
> During this addition, we also noticed a bug in how we calculated the effective MAC operations for both the GRU and EGRU in this DVS gesture recognition task (but not the other tasks), because we were not taking into consideration the effect of input sparsity and inter-layer sparse communication. We have updated the Effective MAC with the correct numbers now — the gap between GRU and EGRU in terms of Effective MAC is slightly larger now than before.
>
> > On a related note in Table 3, I don't see any difference in number of parameters between EGRU and GRU with the same hidden dim in Table 1. Why is there a difference between matched GRU and EGRU model number of parameters in table 3?
>
> We observed that EGRU benefits from larger word-embedding vectors compared to LSTM and GRU. This increases the number of parameters of the word-embedding layer by about a factor of 2. Language models need to compare the output embedding vector via dot product with the embedding vectors of the dictionary. Since EGRU outputs only positive values, we hypothesise that extra parameters are required to cancel terms in the dot-product. We have added this explanation to the supplement.
>
> > What are some of the limitations of the proposed EGRU model? It would be great if the authors could please further highlight room for improvement of EGRU apart from combining with neuromorphic devices in the discussion.
>
> Some of the limitations of the EGRU that we plan to address in future work include (1) The communication using only positive values may be less efficient than using signed values (as we see in the language modelling example); (2) Using an approximate rather than the exact gradient may not be optimal; (3) Since EGRU dynamics are based on the GRU, it does not handle long range dependencies any better than a GRU.
> We will add this in to the discussion in the final version of the paper.

---

> > ### Comment · Reviewer_SZU5 · 2022-11-25
> > **Acknowledging author response**
> >
> > I thank the authors for addressing the comments from my review. It is clear that EGRU is a significant contribution to making GRUs more efficient in terms of # effective MACs and hence running time. Thanks also for raising the issue with the bug in calculating effective MACs for DVS gesture recognition and for addressing the clarification issue of the # of parameters in Table 3. After the authors' response, I am retaining my score as I still strongly feel that the contribution made here will be impactful for training efficient RNN models.

---

### Author Response · Authors · 2022-11-16
**Summary of updates**

We would like to thank all their reviewers for their constructive comments and questions. We have responded to each reviewer separately and in detail. Please note that we have also uploaded an updated version of our main text as well as supplement. We have indicated all major changes using blue color text.

We have added the following in the updated version of the paper:

1. New results for the DVS Gesture Recognition task: Specifically we have added results for GRU+DA and CNN+GRU+DA results and updated the table overall. We have also updated the effective MAC column for this task.
2. A plot of the distribution of neuron activity in the EGRU after training.
3. A discussion of the Identity-RNN paper to the related work.

Moreover, we have fixed all the minor text issues pointed out by the reviewers.

We plan to include the following major additions in the final version of the paper:

1. Limitations of the EGRU in the discussion section.
2. Comparison with different threshold settings and initialisations for the PTB task in the Supplement.
3. A more comprehensive analysis of the effect of network size on sparsity and number of events for the PTB task in the Supplement.

We will also make changes based on the other feedback from the reviewers.

---

### Decision · Program_Chairs · 2023-01-20

**Decision:**

Accept: notable-top-25%

**Justification For Why Not Higher Score:**

Though this paper provides a very nice, clear contribution for making RNNs more energy efficient it is not conceptually or empirically ground-breaking. Thus, although it is worthy of a spotlight, given its high quality, it is probably not appropriate for an oral.

**Justification For Why Not Lower Score:**

The paper is well written and the work very solid. There is no doubt this paper should be accepted. Moreover, given the clear agreement between reviewers that it makes a worthwhile contribution, I think it is worthy of a spotlight.

**Metareview: Summary, Strengths And Weaknesses:**

This paper extends gated recurrent unit (GRU) models by adding an event generating mechanism (making an event-based GRU, or EGRU) that is more sparse and discrete in its activations over time. The authors provide theoretical and empirical data to show that this can permit much more computationally efficient recurrent neural network training without sacrificing performance.

The reviewers all agreed that this is a strong paper that makes an important contribution. There were some concerns related to whether the tests were extensive enough and comparisons to other work sufficient. After the authors' response, the reviewers were satisfied that their concerns were addressed, and there was a clear consensus that this paper should be accepted.

**Note From Pc:**

if the above contains the word "oral" or "spotlight" please see: "oral" presentation means -> notable-top-5% and "spotlight" means -> notable-top-25%. As stated in our emails, we are disassociating presentation type from AC recommendations

**Summary Of Ac-Reviewer Meeting:**

N/A